# Machine learning techniques for continuous genetic assignment of geographic origin of forest trees

**Bernd Degen**[1]*, **Yulai Yanbaev**[2], **Niels A. Müller**[1]

**1** Thünen Institute of Forest Genetics, Grosshansdorf, Germany, **2** Bashkir State Agrarian University, Ufa, Russia

* bernd.degen@thuenen.de

## Abstract

Origin tracking is important to ensure use of the right seed source and trade with legally harvested timber. Additionally, it can help to reconstruct human-caused historical long-distance seed transfer and to spot mislabelling in forest field trials. So far, genetic assignment approaches were mostly discrete, assigning test samples to predefined groups. The main limitation of this approach is the justification of these discrete groups when genetic variation across the landscape is actually continuous. Here, we compare the accuracy of five continuous assignment methods. Specifically, we test a nearest neighbour method (NN), direct gaussian process regression (GPR-D) using the radial basis kernel function, grid based gaussian process regression (GPR-G) applying the Matérn kernel function, genomic prediction (GP) and deep learning (DL), using two genome-wide single nucleotide polymorphism (SNP) datasets of trees from across Europe. The first dataset comprises 30,000 SNPs from 865 European beech (*Fagus sylvatica*) trees, the second dataset consists of 381 SNPs from 1,883 pedunculate oak (*Quercus robur*) trees. The accuracy, as measured by the geographic distance between true and predicted locations, was highest for the GPR-G and DL methods with the beech dataset with a median distance of only 55 km and 76 km, respectively. For the oak data GPR-G and DL also performed best with median distances of 263 km and 278 km, respectively. The relative error (distance/max distance among tree pairs) was below 8% for 90% of all samples for the best method for both datasets. We detected 35 individuals and 10 groups as outliers in the beech data and 27 individuals and 18 groups in the oak data. These outliers may be caused by mislabelling or historical human-caused long distance seed transfer. We discuss the differences in performance of the approaches and highlight future applications and potential for further improvements.

**Data availability statement:** The genotype data, including geographic coordinates of the oak samples, have been deposited in the Dryad Digital Repository and are accessible at the following DOI: https://doi.org/10.5061/dryad.8931zcrrd. All beech sequencing data are publicly available from the National Center for Biotechnology Information (NCBI) Sequence Read Archive (SRA) under the accession number PRJNA1005581: https://www.ncbi.nlm.nih.gov/bioproject/PRJNA1005581.

**Funding:** The work on the oak dataset was supported via a grant from the Waldklimafonds WKF-22WC4111 01 (German Federal Ministry of Food and Agriculture & German Federal German Ministry for the Environment, Nature Conservation and Nuclear Safety), grant No 23RL-1F017 from the Higher Education and Science Committee of the Republic of Armenia in support of the contribution of Yulay Yanbaev. The research on the beech data was funded by a core grant from the Thünen Institute and another grant from the Waldklimafonds WKF-2219WK60A4.

**Competing interests:** NO authors have competing interests Enter: The authors have declared that no competing interests exist.

## Introduction

Most tree species have a broad natural distribution range. In many cases, they are genetically adapted to the different local environmental conditions [1] although the environmental variables and phenotypic traits underlying this process still remain mostly elusive. Local adaptation has been experimentally demonstrated by common garden experiments [2,3]. It can make the selection of appropriate seed sources essential for the success of forest plantations [4] and thus calls for efficient methods to control the declared origin of seed material. For this purpose, genetic methods can play a vital role [5,6].

Several studies have shown historical long-distance transfer of seed material for afforestation. Jansen and Geburek [7] analysed the transnational use of Larch seeds for afforestation in different European countries. The intensive historical transfer of seed material of Norway spruce and Scots pine in Scandinavian countries was reviewed by Myking et al. [8]. Mal-adaptation of the planted trees can be the result of such long-distance seed transfer, especially when the environmental conditions of the seed origin and the later place of use differ strongly [9]. Thus, it would be useful to have a tool that can reconstruct the origin of planted trees that originate from a distant seed source [10]. Genetic control of origin would also be valuable for field trials, as it could help to identify mislabelled plants or errors in the execution of the planting scheme. Moreover, genetic assignment of origin has been applied for timber tracking as a tool to fight illegal logging [11–13]. Finally, ecological genetic studies could benefit from genetic assignment of geographic origins providing information about distances and directions of natural gene flow. Reconstructing the origin of successful pollen and seeds is feasible on smaller geographic scales using paternity analysis and genetic screening with highly variable gene markers e.g., microsatellites [14,15]. However, the precise origin can only be reconstructed if all potential parents in an area have been studied. This is creating an exponentially growing workload with increasing spatial scale. For geographically isolated populations paternity analysis can give minimum distance estimates [16], but still cannot determine the exact origin.

Spatial-genetic assignment methods can be subdivided into discrete and continuous approaches. The distinction aligns closely with the fundamental machine learning terms—classification and regression, respectively. The discrete genetic assignment approaches split the reference samples into groups and try a falsification of the origin of the test samples (named "classification" in machine learning). The assignment is done based on allele frequencies and the likelihood of the genotype of the test individual to occur in the different groups of reference samples [17]. Bayesian approaches have been shown to be more powerful for this assignment [17,18]. Also, a nearest neighbour approach has been proposed as this method is less sensitive to datasets with reference samples of different taxonomical status [19]. Other approaches use clustering methods such as STRUCTURE-like [10] or parameter-free methods like random forests [13]. In the best case, all but one group can be excluded and the geographic area of this group can thus be considered to be the true origin of the test sample.

The continuous spatial-genetic assignment methods used so far predict the expected location of an individual based on the modelling of allele frequencies in a two or three-dimensional space taken from reference samples (called "regression" in machine learning). This approach has been applied to trees [13] but also endangered animals such as elephants [20]. Continuous assignment of origin has also been done with the programs SPASIBA [21] and SPA [22]. SPASIBA stands for Spatial Bayesian Interference and is a further development of the SCAT program [23]. It uses the allele counts of training samples to model the two-dimensional distribution of the allele frequencies. Other than the SCAT software, SPASIBA does not require Monte Carlo simulations but applies Nested Laplace Approximation for the optimisation of the underlying functions. The spatial ancestry analysis (SPA) also models allele frequencies as continuous functions in geographic space but using logistic functions optimized by Newton's based methods.

The advances in next generation DNA sequencing provide much larger genetic datasets that are now also available for studies on geographic origin. In addition, machine learning approaches are progressively used to analyse these data. In forestry, both developments have been merged for genomic predictions (GP) of phenotypic traits important for tree breeding [24,25]. Also, Gaussian process regression (GPR) is a modern machine learning approach that has recently been applied to predict the origin of timber using stable isotopes and trace elements [26], and to reconstruct ancient migration routes of humans based on whole genome data [27]. Maldonado et al. [28] used different machine learning models to assign the genetic sub-population in a Eucalyptus progeny-provenance trial based on foliar spectral information as features and SNP-data to define the sub-populations (output). Recently, deep learning (DL) has also been applied to assign the origin of horse breeds using data of a large SNP array [29].

In the present paper we test and compare the power of a nearest neighbour approach (NN), genomic prediction (GP), gaussian process regression (GPR) and deep learning (DL) using SNP data of georeferenced tree samples from across Europe to predict the geographic origin of trees (continuous spatial-genetic assignment). Further, we identify outliers with large differences between true and predicted geographic origin, which may be due to long-distance seed transfer or mislabelling within trial sites.

## Materials and methods

### Genetic data

**Beech.** We used a subsample of 30.000 polymorphic SNPs from the whole-genome data of 865 *Fagus sylvatica* trees [30]. The SNPs were randomly sampled over all chromosomes. The trees represent 6 to 10 individuals from 99 different beech provenances distributed across the natural distribution range (Fig 1).

**Oaks.** For the oaks, 381 polymorphic SNPs (359 nuclear, 17 chloroplast and 5 mitochondrial SNPs) were taken from 1,883 trees from 188 locations with 6 to 20 individuals per location [32] sampled over a large part of the natural range (Fig 2). The chloroplast and mitochondrial SNPs were maternally inherited in oaks (gene flow only via seeds). In contrast the nuclear SNPs have a bi-parental inheritance and are more broadly dispersed via pollen and seed. The nuclear SNPs were developed by high coverage ddRAD-sequencing [33] and the plastid makers were developed by low coverage whole-genome sequencing. The location of these markers in the chloroplast and mitochondria was confirmed by mapping against the reference plastid sequences [34]. We only took samples that were located inside the natural range [31]. All of the trees had an admixture coefficient for *Quercus robur* of more than 0.8. The admixture was estimated with the program STRUCTURE based on the nuclear SNPs [35].

### Statistical analysis

The statistical analyses were done with R [36] and custom scripts written in Python (version 3.12).

### Data structure and pre-processing

The data matrix X represents the genetic information used for the geographic assignment. It consists of rows corresponding to individuals and columns corresponding to single nucleotide polymorphisms (SNPs). Each element in the matrix

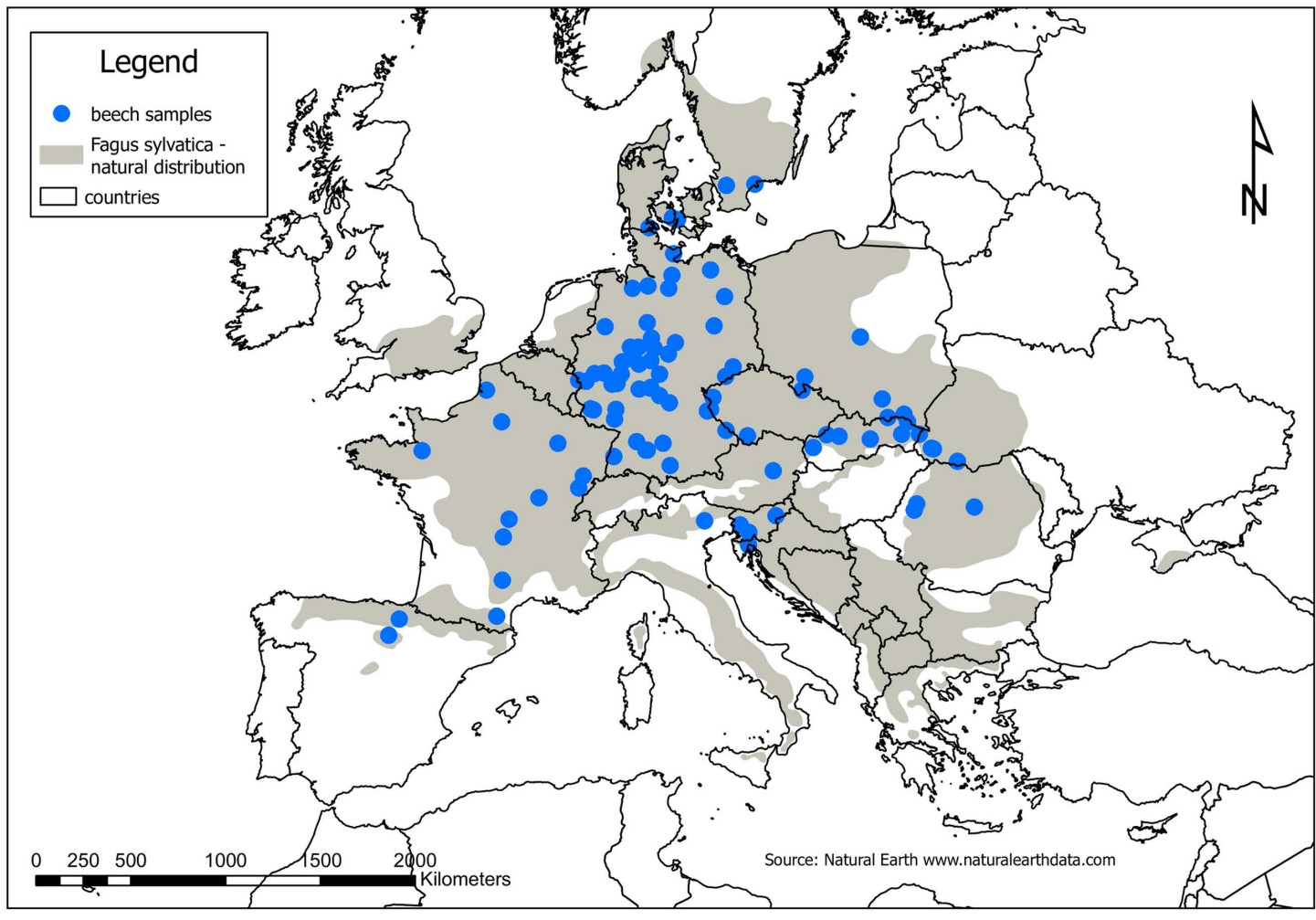

**Fig 1. Distribution of sampled trees and natural range of *Fagus sylvatica* [ 31], shapefile of country borders from www.naturalearthdata.com.**

encodes the genotype of an individual at a given SNP locus. Rows ($n$): Each row represents a single tree individual ($i = 1$, $2,…,n$), where $n$ n is the total number of sampled individuals. Columns ($p$): Each column corresponds to a specific SNP marker ($j = 1,2,…,p$), where $p$ is the total number of SNPs included in the analysis.

The selected individuals and the selected SNPs had less than 5% missing data. The genotypes at the SNPs were transformed to 1 (homozygote reference allele), 0.5 (heterozygote) and 0 (homozygote for alternative allele) for the nearest neighbour approach (NN) and for the gaussian process regression grid approach (GPR-G). They were transformed to 1 (homozygote reference allele), 0 (heterozygote) and −1 (homozygote for alternative allele) for the other three machine learning algorithms. For the nearest neighbour approach (NN) and the genomic prediction (GP), missing values were coded as "NA" and for the two types of gaussian process regression (GPR-D, GPR-G) and deep learning (DL) missing values were imputed using average allele values of all other individuals for that particular SNP. The imputation method should only have a very limited influence on the results because of the low amount of missing data. For the GPR-D, GPR-G and DL only SNPs were used that had a statistically significant correlation either with longitude or latitude. After applying a p-value threshold with Bonferroni correction, 4,335 out of the 30,000 SNPs were selected for GPR and DL analysis of the beech dataset, and 223 (203 nuclear SNPs, 16 chloroplast and

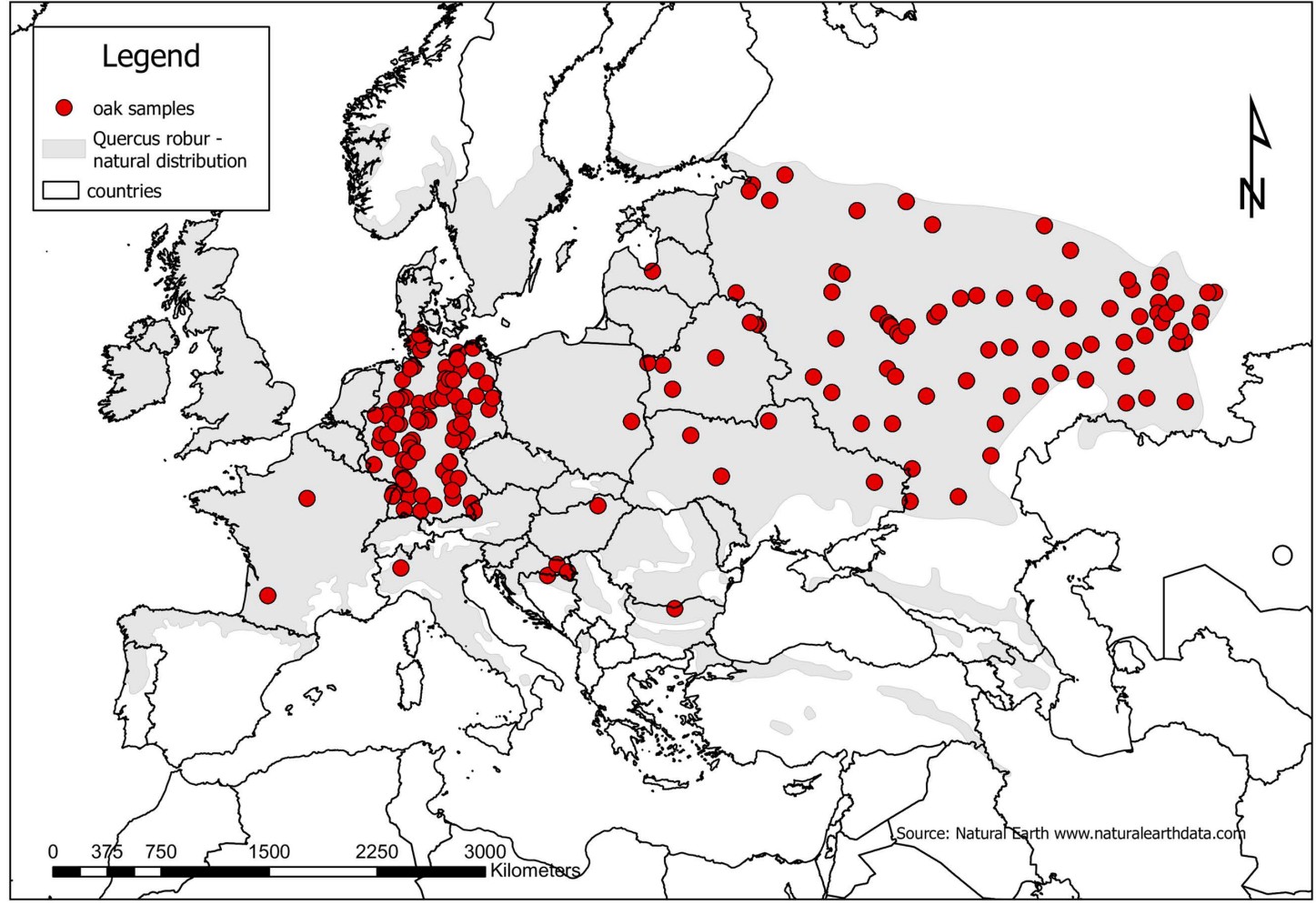

**Fig 2. Distribution of sampled pedunculate oak trees and natural distribution range of *Quercus robur* [ 31], shapefile of country borders from www.naturalearthdata.com.**

4 mitochondrial SNPs) out of 381 SNPs for the oak dataset. The plastid SNPs of the oaks were treated in the data processing as homozygotes.

### Nearest neighbour approach (NN)

Following Degen et al. [19] we computed the genetic distance (mean absolute difference of allele frequencies) between all pairs of trees and used this to generate a continuous assignment of the mean latitude and longitude of the *k* genetically most similar individuals as the predictor. We tested different k values (5, 10, 15, 20) but got the highest accuracy in the cross-validation for k = 5 (see below), which was then applied for both datasets.

### Genomic prediction (GP)

We utilized the genomic Best Linear Unbiased Prediction (gBLUP) algorithm to compute the predicted latitude and longitude for each tree. For that we applied the "kin.blup" function of the "rrBLUP" R package [37]. An essential part of gBLUP is the construction of a kinship matrix. This matrix quantifies the genetic similarity between pairs of individuals based on

their allele profiles at the SNPs and thus the kinship matrix provided estimates of co-variation between individuals for the predicted target values (latitude, longitude).

## Gaussian Process Regression- direct approach (GPR-D)

We implemented GPR using the GPyTorch library in Python. The GPR model was formulated with a constant mean function to represent the mean of the Gaussian process and the Kernel Function. A Radial Basis Function (RBF) kernel was chosen for its ability to model smooth functions. The kernel had the form: $k\left(x_i, x_j\right) = \sigma^2 \exp\left(-\frac{\|x_i - x_j\|^2}{2l^2}\right)$, with $k\left(x_i, x_j\right)$ representing the covariance function or kernel function. It measured the similarity between two input points $x_i$ and $x_j$, in our case the allele composition of individuals across loci (genotypes). The kernel function determines the shape of the functions that the Gaussian process can model. In that $\sigma^2$ is the variance parameter. It scales the overall magnitude of the kernel. The exponential function $\exp\left(-\frac{\|x_i - x_j\|^2}{2l^2}\right)$ ensures that the kernel function produces values between 0 and $\sigma^2$.

The term inside the exponential function controls how rapidly the covariance decreases with distance between points. The term $l$ is the lengthscale parameter. Larger values of $l$ imply a smoother function, where points farther apart in the input space are still considered similar.

An optimization of hyperparameters was done using a random search approach. A total of 10,000 random configurations of hyperparameters were evaluated. Each configuration included:

• lengthscale_lat: Lengthscale for the latitude predictions.

• noise_lat: Noise level for the latitude predictions.

• lengthscale_lon: Lengthscale for the longitude predictions.

• noise_lon: Noise level for the longitude predictions.

The random values were sampled uniformly within the following ranges:

• Lengthscales: [0.001, 400]

• Noise levels: [1e-4, 1]

For each hyperparameter-configuration, a cross-validation was performed. For this the data was randomly split, with 90% used for training and 10% for testing in each iteration. This was repeated five times. The mean haversine distance between the predicted and actual locations was used as the evaluation metric. The GPR was used in this direct approach to train two models using the entire allele profiles of the training individuals: one to predict latitude and one to predict longitude.

## Gaussian Process Regression- grid approach (GPR-G)

In the grid-based Gaussian Process Regression (GPR-G), we again used the GPyTorch library in Python but with the Matérn kernel function and a parameter $\nu = 2.5$ that controlled the smoothness of the function. The study area was divided into a fixed grid with each cells measuring 20 km x 20 km for the beech data and 50 km by 50 km for the oak data. The grid spanned a latitude and longitude range from the minimum to the maximum value observed among our individuals. The allele frequencies for each SNP were pre-aggregated for all training individuals located within each grid cell. A different GPR model was trained to predict allele frequencies for each SNP across the grid cells. This means for beech 4,335 and for the oaks 223 different GPR models were created. Hyperparameters were not explicitly tuned. Instead, the model parameters were optimized over 10 epochs by minimizing the marginal log likelihood of the training data with the Adam optimizer. In each grid cell we used the predicted allele frequencies at all SNPs to compute the likelihood to obtain the

genotype of the test individual. The underlying assumption was that genotypes occur as expected in a Hardy-Weinberg Equilibrium (HWE). The grid cell with the highest probability was selected as predicted location (maximum likelihood approach).

### Deep learning (DL)

We employed a feedforward deep neural network implemented in the R package "H2O" [38] to predict latitude and longitude from genetic data. The H2O cluster offers optimized parallel processing, enhancing computational speed. The model architecture consisted of five hidden layers with the following neuron configuration: 100, 100, 100, 50, and 50 neurons, respectively. We used the "RectifierWithDropout" activation function with dropout ratios of 0.1 for the input layer and varied dropout ratios of 0.3, 0.3, 0.3, 0.2, and 0.2 for the hidden layers [39]. L1 and L2 regularization (1e-6) were applied to mitigate overfitting. Dropout is a technique where a proportion of neurons are randomly set to zero during training, which helps in preventing overfitting and improving generalization by reducing dependency on specific neurons. The model was trained for up to 500 epochs with early stopping based on Root Mean Square Error (RMSE) and a tolerance of 0.001. RMSE measures the average magnitude of errors between predicted and actual values. The training process halts if RMSE does not improve by at least 0.001 for ten consecutive rounds. To ensure robust predictions, an ensemble of five models was used. The outputs of these models were aggregated to compute the mean and variance of the predictions (ensemble learning).

### Cross validation – leave-one-out approach

We assessed the accuracy of the NN, GP, GPR-D, GPR-G and DL analyses using a leave-one-out cross-validation approach. This was done by a loop over all individuals treating each individual in the dataset as the test data while using the remaining individuals as training data. For this, the GP analysis set the true values for latitude and longitude of the individual under test temporarily to "NA" (to simulate unseen data). In the NN, GPR-D, GRG-G and DL analysis, all individuals except the one being tested (test data) were used as training data. The predicted latitude and longitude values obtained from these analyses were then compared with the true values, i.e., the passport data. We calculated the spatial distance between the predicted and actual locations with the Haversine formula [40]. The formula computes the shortest distance over the earth's surface between two geographical points. As another measure of accuracy, we computed the Pearson's correlation coefficient between given and predicted latitude as well as longitude values.

### Identification of outliers

In each of the five methods we computed the standard deviation or variance for the predicted location. The standard deviation of the model predictions is directly computed by the Gaussian Process Regression (GPR) models. The function for the genomic prediction (GP) also provides an error estimation called "prediction error variance" (PEV). The "kin.blup" function used the mixed model equations and the genomic relationship matrix to compute the PEV for each prediction. Same as for GPR these variance values were used to computed the upper limit of the 95% distance interval. For the nearest neighbour approach (NN) we computed mean and standard deviation of the pairwise distances between the five identified nearest neighbours, and for the deep learning (DP) we computed the pairwise distances between the predicted locations of five repetitions. Based on these standard deviations we calculated the 95% upper limit for the distance radius. We searched for outliers among the individuals but also for outliers of entire groups. For the group approach the means of all predicted latitude and longitude values as well as the means of the 95% upper limits were computed. The distribution of distances was used for an interquartile range (IQR) methodology to identify outliers in the predicted locations. This approach computes as range the difference between the 25th and 75th percentile of the quotient distribution (IQR). Our application used the 75th percentile + 1.5 times the IQR (1.5 x IQR) as threshold for the group outliers and the 75th percentile + three times the IQR (3 x IQR) as threshold for the individual outliers. The IQR approach does not assume normal

distributed values and is robust to extreme values. In both approaches only those outliers were kept for which the distance between given and predicted location was larger than the 95% upper limit for the distance radius.

## Results

### Pairwise geographic distances

The pairwise geographic distances between the beech samples (Fig 3a) had a mean of 621 km (standard deviation: 395 km). The oak samples covered a much larger area with a mean pairwise geographic distance between trees of 1452 km and a standard deviation of 1030 km (Fig 3b). These distributions are important to assess the accuracy of the used assignment methods.

### Accuracy of predicted locations

For the beech dataset, the grid based Gaussian process regression (GPR-G) and deep learning (DL) gave the highest accuracies with median distances of 55 km and 76 km, followed by the nearest neighbour (NN) method, direct Gaussian process regression (GPR-D) and genomic prediction (GP) (Table 1). Notably, the standard deviations of NN and GPR-G were larger than those of the other three methods. Based on the Pearson's correlation between given and predicted latitude and longitude values, DL and GPR-G performed best with mean values of 0.97 and 0.95 (Table 2). The ranking of methods is also visible by the curves of the relative assignment errors, calculated as the distance between true and predicted location divided by the maximum distance between two data points (Fig 4). Here, the good performance of DL and

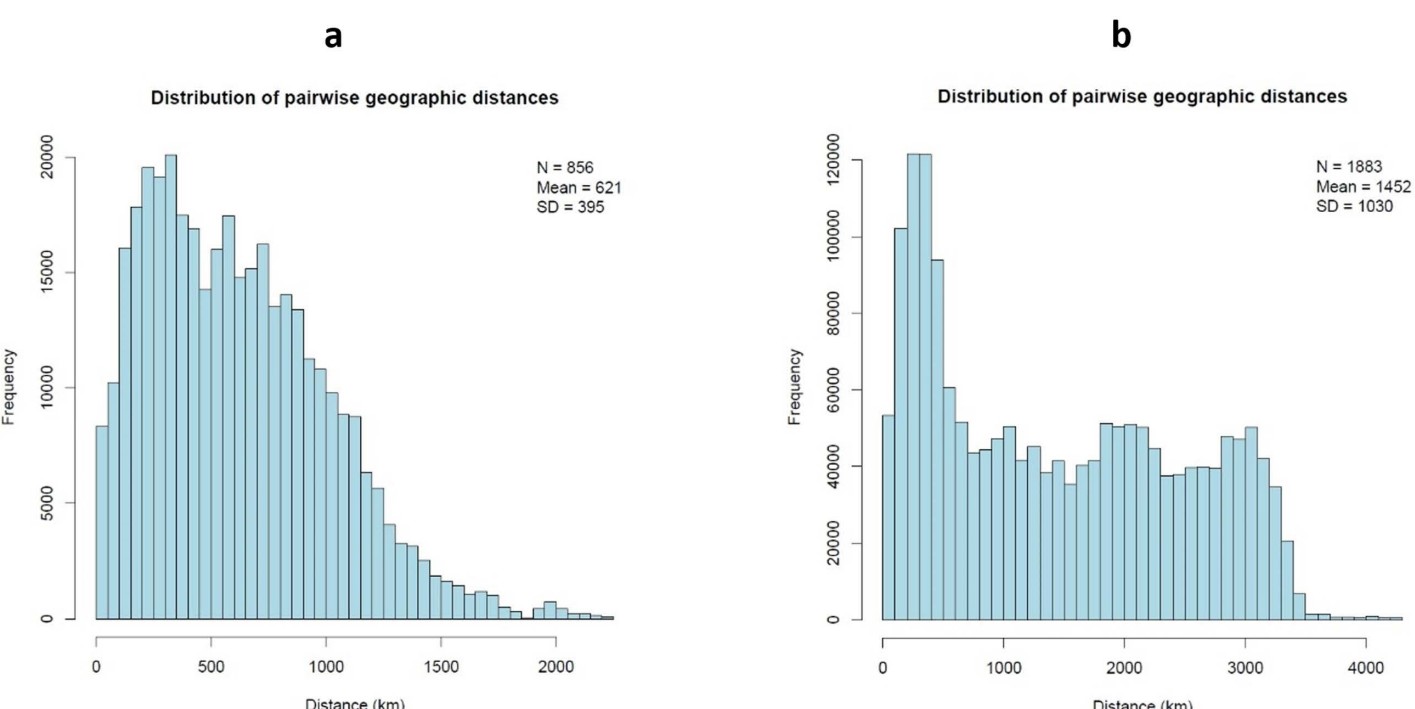

**a**                                                                                        **b**

**Fig 3. Histogram of pairwise distances between samples.** (a, b) The y-axis shows the frequency and the x-axis the geographic distance in km of all pairwise comparisons between samples in beech (a) and oak (b). The number of samples (N), their mean pairwise distance and the standard deviation (SD) of that distance are given in the top right corner.

**Table 1. Results of the leave-one-out cross validation for the two datasets "Fagus" and "Quercus" on the accuracy of the predicted locations measured as distance between true and predicted locations.**

| Dataset | Method | Mean distance (km) | Median distance (km) | SD Distance (km) | N outliers individuals | N outliers groups |
|---------|--------|--------------------|----------------------|------------------|------------------------|-------------------|
| Fagus | DL | 107 | 76 | 100 | 12 | 4 |
| Fagus | NN | 135 | 97 | 130 | 4 | 0 |
| Fagus | GPR-D | 154 | 141 | 97 | 5 | 6 |
| Fagus | GPR-G | 98 | 55 | 122 | 29 | 5 |
| Fagus | GP | 147 | 127 | 96 | 2 | 0 |
| Quercus | DL | 312 | 278 | 210 | 7 | 4 |
| Quercus | NN | 365 | 281 | 300 | 8 | 3 |
| Quercus | GPR-D | 356 | 295 | 243 | 3 | 8 |
| Quercus | GPR-G | 363 | 263 | 343 | 10 | 10 |
| Quercus | GP | 366 | 312 | 238 | 6 | 10 |

**Table 2. Pearson's correlation coefficient between given and predicted latitude and longitude values for the five methods.**

|  | Latitude | Longitude | Mean |
|--|----------|-----------|------|
| **beech** |  |  |  |
| DL | 0.964 | 0.978 | 0.971 |
| NN | 0.915 | 0.943 | 0.929 |
| GPR-D | 0.891 | 0.958 | 0.924 |
| GPR-G | 0.937 | 0.963 | 0.950 |
| GP | 0.917 | 0.959 | 0.938 |
| **oaks** |  |  |  |
| DL | 0.874 | 0.983 | 0.928 |
| NN | 0.670 | 0.945 | 0.808 |
| GPR-D | 0.614 | 0.960 | 0.787 |
| GPR-G | 0.615 | 0.945 | 0.780 |
| GP | 0.656 | 0.957 | 0.807 |

GPR-G is also visible with the lowest relative error distribution. The correlations between the distances of most methods were only moderate with values between 0.35 and 0.61 (Table 3). Higher correlations were identified between GP and GPR-D (0.77) and between DL and GP (0.72).

Also, in the oak dataset with fewer SNPs, GPR-G and DL performed best, with a median distance of 263 km and 278 km (Table 1). However, the other methods showed similar performance with median distances of 281 km for NN, 295 km for GPR and 312 km for GP (Table 1). The standard deviation of distances was smaller for DL but relatively high for GPR-G compared to the other methods. In the oak dataset the mean Pearson's correlations between given and predicted locations were generally weaker compared to the beech dataset, especially the correlations for latitude (Table 2). Based on these criteria, DL with a mean correlation of nearly 0.93 performed better than the other four methods. There was a high Pearson's correlation coefficient of 0.92 between the distances of GPR-D and GP (Table 4). All other correlations were equal or smaller compared to the ones from the beech data. The relative errors (Fig 4b) showed the good performance of DL but an ambivalent performance for GPR-G with very low errors for about 50% of the cases but higher errors compared to DL for the other 50% of samples.

It should be noted that the best methods, that is DL and GPR-G, for the beech dataset both had sample errors below 7.9% for 90% of the samples and for the oak dataset this was DL with an error of less than 8.5% for 90% of the samples.

**A**

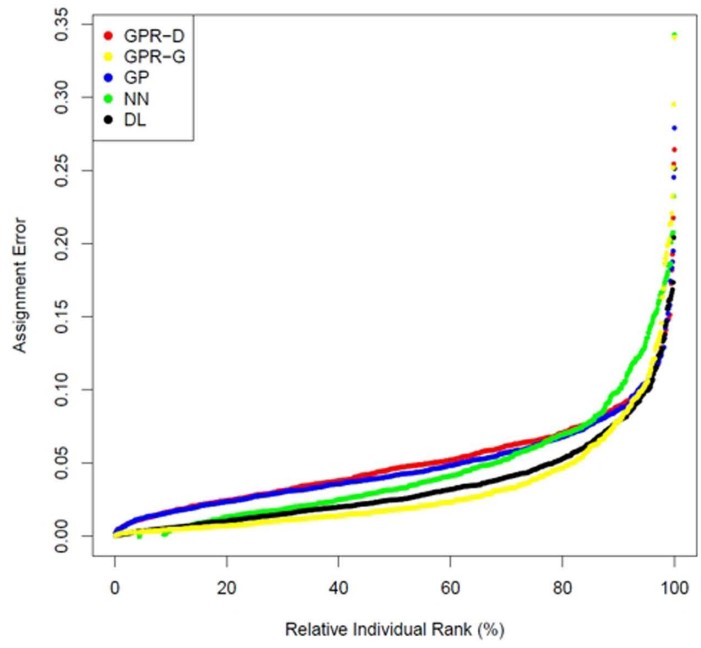

**B**

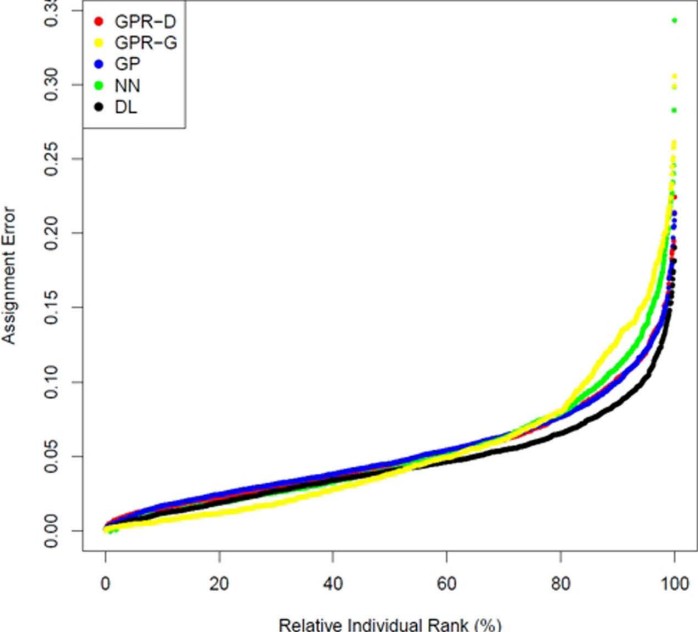

**Fig 4. Assignment errors estimated using a dataset of beech (A) and oak (B).** The errors are expressed as fraction of the distances between true and predicted location from the maximum distance among individuals. For each approach the individuals are ordered from left to right according to the error.

**Table 3. Pearson's correlation coefficient between distances between given and predicted locations for the five methods for the beech dataset.**

|  | DL | NN | GPR-D | GPR-G |
|---|---|---|---|---|
| NN | 0.350 |  |  |  |
| GPR-D | 0.615 | 0.465 |  |  |
| GPR-G | 0.425 | 0.540 | 0.504 |  |
| GP | 0.721 | 0.533 | 0.776 | 0.562 |

**Table 4. Pearson's correlation coefficient between distances between given and predicted locations for the five methods for the oak dataset.**

|  | DL | NN | GPR-D | GPR-G |
|---|---|---|---|---|
| NN | 0.336 |  |  |  |
| GPR-D | 0.467 | 0.546 |  |  |
| GPR-G | 0.275 | 0.432 | 0.478 |  |
| GP | 0.437 | 0.532 | 0.918 | 0.447 |

### Outliers with large distance between given and predicted geographic origin

For the beech dataset a total of ten group-outliers were identified out of 99 groups (between 0 and 6 per method, Table 1). The methods GP and NN did not identify outlier groups. The same provenances 32, 111 and 9 were identified by the three methods DL, GPR-G and GPR-D (Sup 1 Fig). In each case the provenance 32 with a given location in North-East Germany had group predictions much more in the South-West of Germany, the provenance 111 with given location in the West of Czechia got predicted positions much more in the East (Slovakia or Northern Hungary). The provenance 9 in North-West France had predicted locations more in the south-eastern part of France. The other seven group-outliers were obtained only with one method (Sup 1 Fig). A total of 35 individuals (2–29 per method) were identified as outliers (Sup 2 Fig). One individual, B9 of provenance 51 was identified as outlier by all five methods and likely represents a planting error. The given location in West Germany was in all cases projected a few hundred kilometres more to the East. The individual H65 of provenance 32 was identified by four methods as outlier, two individuals were identified by three methods (E33 from provenance 32 and F75 from provenance 101). Individual F75 had a given location in South Germany but was predicted to be located much more in the North of Germany. Six individuals were identified as outlier by two methods and the remaining 28 just by one method. In most cases the individual outliers belonged to groups that were also entirely identified as an outlier (Sup 1 Fig, Sup 2 Fig).

For the oak data the different methods identified between three and ten outlier-groups, summing up to a total of 18 different groups out of 188 groups assigned as outlier (Table 1, Sup 3 Fig). The location 213 in Latvia was identified by all five approaches as an outlier-group with a predicted location much more in the South-East in West Russia. Four methods spotted location 210 (Ukraine) to be from a region more North-East in West Russia. Three outlier-groups were identified by three methods (437 Russia, 5 France, 219 Croatia). Here, samples from the French location were predicted to have an origin in West Germany, the Croatian samples were supposed to be from a region much more in the North-East and the samples from location 437 in Russia are assigned to a region much more in the West of Russia. All other group-outliers were identified by only two methods (4) or a single method (9). On the individual level between three and ten outliers were identified per method, summing up to a total of 27 individuals (Table 1, Sup 4 Fig). Most of the individual outliers were from locations that have also identified as a group outlier (Latvia 213, Russia 437, Croatia 219, Ukraine 210, France 5) but also a few that were not spotted on the level of groups were identified. Examples for this are samples from location 233, 236 and 238 (Germany) that were predicted to be from a much more eastern origin.

## Discussion

### Performance of different continuous assignment methods

The two datasets for beech and oak differed largely in the spatial scale of the covered area. The oak samples are distributed across an area from France to Siberia which is more than two times larger than the region covered by the beech samples. Thus, it is useful to generate a relative measure for the error to compare the accuracy among methods, datasets and different studies. We followed the suggested approach of Guillot et al. [21] and computed the relative error as the distance between passported and predicted distances in relation to the maximum distance between two samples in the data set.

The grid-based gaussian process regression (GPR-G) and deep learning (DL) performed best for the beech data with a median difference between passport and predicted location of 55 and 76 km (error = 1.8 and 2.4%) and for the oak data with 263 and 278 km (error = 3.8 and 4%). But DL performed better than GPR-G if we consider also the correlation between given and predicted latitude and longitude values. The differences in mean correlation between the two methods were small for the beech data set (0.97 versus 0.95) but larger for the oak data set (0.93 versus 0.78). The nearest neighbour approach (NN) was on the third rank in both cases based on the median but had also a large proportion of individuals with large distances. The other two approaches genomic prediction (GP) and direct gaussian progress regression (GPR-D) performed slightly worse, although differences of relative errors between methods were minor especially for the oak data.

It may not be surprising that DL was the best method for both data sets. Deep learning is known to be highly suitable for large-scale studies where complex pattern of genetic markers need to be captured [41,42]. DL models can automatically learn from the patterns associated with different geographic origins. Not much user input in terms of parameter tuning is needed for the implementation of DL in the computing environment H2O [38]. The disadvantages of DL were the higher requirements for computing resources and the difficulty to interpret the "black-box" neural network [43]. We tested if we can increase the precision of the predicted origin by further enlarging the complexity of the neural network but there was no significant improvement neither by increasing the number of layers nor by using more neurons per layer.

The performance of the two variations of the gaussian process regression was quite different. The grid-based approach (GPR-G) with aggregation of allele frequencies of all individuals in the same grid cell, application of the Matérn kernel function, modelled allele frequencies and a maximum-likelihood approach for test genotype origin was clearly better than the direct prediction (GPR-D) of latitude and longitude based on the complete genotypes. The grid-wise approach is similar to the approaches implemented in the SCAT and SPACIBA programs discussed below. Our direct approach without the intermediate step of allele frequency modelling has not been used so far. The weaker performance of GPR-D can be explained by the less suitable radial kernel function compared to the Matérn kernel function. The Matérn kernel function is known to be good for modelling spatially autocorrelated data often found in ecology research [26].

The strongest correlation between methods was observed in both datasets for GP and GPR-D which can be explained by the fact that they are both picking up the same pattern and using a kernel functions for the predictions (although the kernel function it linear for GP).

So far, genomic predictions or genomic selection have been mostly used in breeding programs to estimate breeding values of individuals [44]. This was first done for animal breeding, followed by crop breeding and since a few years also tree breeding [24]. There have been many different algorithms proposed to compute breeding values or to predict phenotypes [45]. Among them genomic best linear unbiased prediction (gBLUP) is widely used. For the predictions with gBLUP the co-variation between kinship-similarities and similarities of phenotypes is used. In breeding programmes, the success of this approach is linked to the complex genetic architecture of the predicted traits that are highly polygenic with small effects of each causal allele that is well captured with a kinship-matrix. In our application "the complex genetic architecture" of the traits "longitude" and "latitude" is generated by different processes and the involved genetic variants causing spatial genetic structures at different spatial scales. Among them are demographic processes affecting the whole genome,

especially the recolonisation from the last glacial refugia [46] in combination with limited pollen and seed dispersal creating an isolation by distance pattern [47,48] but also other processes that only impact parts of the genome such as local adaptation [30,49,50] or spatially differential introgression [51,52].

The nearest neighbour approach (NN) is a non-parametric method used for classification and regression. Among the used methods the calculation of NN is the fastest one because it is just an averaging of latitude and longitude values of the k nearest neighbours. The k nearest neighbours have been identified based on genetic similarity to the tested individual. The method has been successfully used for discrete geographic assignment of different tropical tree species genotyped with different gene markers (SNP and microsatellites): Sapelli [19], trees of the genus *Dipteryx* [53], and *Hymenaea* [54]. In these cases, NN performed better compared to Bayesian methods because it could pick up complex pattern in the reference data caused by a mixture of genotypes from related taxa or a mixture of individuals from different postglacial recolonisation routes. A clear disadvantage is that NN cannot predict longitude and latitude values outside the range of the reference data and it is proposed to be sensitive for non-representative sampling of reference data.

An open question is to which extend the selection of the species has an impact on the accuracy of the machine learning based prediction of genetic origin. Generally, the accuracy is linked to the extent of genetic differentiation among populations and more explicitly to the spatial autocorrelation of the genetic data used as input features. Recently, Milesi et al. [55] calculated in a comparative study with large sets of SNPs of seven European tree species for *Fagus sylvatica* and *Quercus petraea* (a related species to *Quercus robur*) intermediate measures of genetic differentiation ($F_{ST}$ = 0.05, 0.04) whereas other species such as *Pinus pinaster* and *Populus nigra* had much stronger values of genetic differentiation (0.13, 0.16) and the species *Pinus sylvestris* and *Betula pendula* only very small $F_{ST}$-values (0.01, 0.03).

Also, the sample design has an impact on the precision of the geographic predictions. We assume a systematic sampling scheme with sampled locations matching grid points distributed over the whole species range would be ideal. Each sampled location should be represented by three to ten individuals. The distances among neighboured grid points are limited by the available budget for sampling and genotyping.

## Other studies on continuous assignment

One of the first continuous assignment of geographic locations based on genetic data has been done by Wasser et al. [23] with the SCAT program. They used genetic data of 16 microsatellites from 399 elephant samples collected all over the natural distribution range in Africa. In their spatial smoothing approach, they modelled the allele frequencies in a two-dimensional space. The distribution of allele frequencies followed independent Gaussian processes and were allowed to vary in a spatially correlated way. They used Monte Carlo simulations for the optimisation of the underlying functions. Their median distance between predicted and true location of the elephants was 499 km. That corresponds to a relative error of 7.9%. Guillot et al. [21] introduced the program SPASIBA, which stands for Spatial Bayesian Interference and is a further development of the SCAT program. It also uses the allele counts of training samples to model the two-dimensional distribution of the allele frequencies. Other than the SCAT software it does not require computing intense Monte Carlo simulations but applies nested Laplace approximation for the optimisation of the underlying functions. Two examples of SPASIBA predictions of individual locations were given by Guillot et al. [21]. First, they predicted the bird origin of the Florida scrub jay based on 41 SNPs. The median distance was 26.4 km corresponding to a relative error of 9.1%. Second, they predicted the origin of *Arabidopsis thaliana* in Europe using genetic data of 1007 individuals at 1000 SNP loci. The distance at the 75% percentile was 93 km corresponding to a relative error of less than 2%.

The SCAT program was also used by Puckett and Eggert [56] to predict the geographic origin of American black bear (*Ursus americanus*) based on 1000 SNPs and 15 microsatellite loci. The accuracy ranged between 192−902 km (error 3.4%−16.4%) depending on the gene maker set used. Further, the SPASIBA program was used by Finch et al. [13] to predict the geographic origin of Spanish Cedar (*Cedrela odorata*) in Latin-America. They used 140 SNPs for 386 individuals and came up with a median distance of 189 km between true and predicted location (error = 4%).

Another modelling approach named SPA for continuous assignment has been proposed by Yang et al. [22]. The spatial ancestry analysis (SPA) also models allele frequencies as continuous functions in geographic space but using logistic functions which were optimized by Newton's based methods. Unfortunately, both programs SPASIBA and SPA were not updated anymore and we could not apply them to our oak and beech data. Nevertheless, the supplementary material of Guillot et al. [21] contains results on comparative application of SPASIBA and SPA for a data set of *Arabidopsis thaliana* from Horton et al. [57]. These data are available and have genotypes at 1000 SNPs, sampled from a total of 215 k variants, for 1107 individual distributed over Eurasia. In a comparison we used this data with our five methods. We compared the accuracy of the predicted geographic locations of our five methods (GPR-D, GPR-G, GP, NN, DL) with results obtained by the programs for continuous assignment SPASIBA and SPA (supplementary 5). The four machine learning approaches (GP, GPR-D, GPR-G, DL) provided errors between the results of SPA and SPASIBA. The median error of NN and GPR-G was even better than that of SPASIBA.

Mortier et al. [26] used Gaussian process regression and the Matérn kernel function to predict the origin of timber in East-Europe based on data for stable isotopes and trace elements. In frame of a four-fold cross-validation they observed average distances between true and predicted location of 179 to 228 km depending of the tree species.

## Speciality of our approaches

There is an important difference between four of our methods and the above-mentioned examples. For NN, GPR-D, GP and DL we used a direct *prior* probability approach to predict locations and the others applied *posterior* probability approach. The four methods focused on predicting geographic coordinates directly from genetic data without relying on intermediate steps such as allele frequency modelling or computing *posterior* probabilities across different grid cells. But our GPR-G method matches with the above cited other approaches used in SPASIBA, SCAT and SPA.

Another direct approach has been done by Battey et al. [58] using the deep learning network named "Locator" for geo-referenced whole genome data of *Plasmodium parasistes*, *Anopheles mosquitoes* and global humans to predict geographic origin. They tested architectures of different complexity for the predictions and finally used a network with ten layers and 256 nodes each for runs with GPUs. Our network was less complex (5 hidden layers, three with 100 and two with 50 neurons). Their error rates were below 1%. They trained their network with 90% of the samples running repetitions along windows in the genomes of 500kbp to 10Mbp. The variation between the windows served as estimator of uncertainty. We used all genetic data but repeated the network training five times to gain an estimator of uncertainty.

## Sampling design in future studies

The effectiveness of continuous genetic assignment methods depends heavily on the structure of the reference dataset. For future studies with the objective to capture the spatial genetic structure, a critical choice has to be made between individual-based sampling and population-based sampling. In the individual based approach, individuals are sampled across the landscape in a spatially uniform manner to ensure a broad geographic coverage. This is particularly useful when genetic variation follows a continuous gradient (i.e., isolation by distance) and there are no discrete populations [48,59]. Individual-based sampling improves the precision of spatial assignment models, especially for machine learning approaches like Gaussian Process Regression (GPR-G) and Deep Learning (DL), which benefit from fine-scale spatial data [58,60]. In contrast, population-based sampling focuses on well-defined populations. Here, multiple individuals are collected per site but with fewer total locations. This approach is useful for species with strong population structure due to historical refugia, geographic barriers, or limited gene flow [61]. This sampling approach is particularly suited for discrete assignment methods, such as Bayesian clustering or STRUCTURE-like models. However, when genetic variation is not strongly clustered but varies continuously over space, population-based sampling can introduce bias, leading to an overestimation of differentiation between populations and an underestimation of within-population variability [59,62].

## Practical implications of the identified outliers

The identified outliers in the study have practical implications for forest management and genetic research. First, these outliers highlight potential cases of long-distance seed transfer or mislabelling. These issues are critical to ensure the accuracy of provenance trials and the selection of future seed harvesting stands. For example, detecting outliers where the predicted and actual locations of tree origins differ substantially may indicate historical human interventions in seed distribution (translocations or unintentional admixture). This information is elementary for future seed sourcing strategies and may help to correct past errors in seed origin documentation. Special importance is given for tree species and regions with known human-mediated seed transfer such as Larch seeds used for afforestation in different European countries [7], seed material of Norway spruce and Scots pine in Scandinavian countries [8] or for introduced exotic tree species with a huge natural distribution range on the continent of origin such as red oak (*Quercus rubra*) or Douglas fir (*Pseudotsuga menziesii*) in Europe [10].

Genetic timber tracking as law enforcement tool to fight illegal logging [11] can also benefit from our new approaches of continuous prediction of tree origin. For timber harvested in natural forests or forests generated with local seed sources, differences between predicted and passported origin (outliers) can indicate misdeclarations and uncover fraud. In addition to former categorial classification approaches the continuous assignment provides useful information on true origin and thus potential locations of possible crime.

Moreover, the ability to identify outliers combined with information of growth and viability of the outliers can serve as a powerful tool for monitoring the adaptation and survival of tree species in changing climates. Trees that are significantly out of place might exhibit maladaptation, which can be detrimental to their survival. Therefore, identifying and understanding the reasons behind these outliers can lead to more effective strategies for selecting of future seed sources that are better suited to current and future environmental conditions.

Furthermore, our findings underscore the importance of using advanced continuous genetic assignment methods, such as deep learning, genomic prediction and Gaussian process regression, which have proven effective in identifying geographical origins and outliers. As the study shows, applying these methods can improve the accuracy of geographic origin predictions by avoiding definition of groups, making them invaluable for forest genetic management, conservation efforts, and law enforcement related to seed trade, timber tracking and illegal logging.

## Supporting information

**S1 Fig. Maps showing the group outliers identified in the beech dataset.**
(DOCX)

**S2 Fig. Maps showing the individual outliers identified in the beech dataset.**
(DOCX)

**S3 Fig. Maps showing the group outliers identified in the oak dataset.**
(DOCX)

**S4 Fig. Maps showing the individual outliers identified in the oak dataset.**
(DOCX)

**S5 "Supplementary 5.docx". Comparison of prediction accuracy with previously reported methods.**
(DOCX)

**S6 "Supplementary 6.zip". Python programs, r-scripts and input data for the five continuous geographic assignment methods.**
(ZIP)

## Acknowledgments

We thank two anonymous reviewers for their helpful comments on a former version of the manuscript and we are thankful to Malte Mader for his support in using the High-Performance Computing Cluster of the Thünen-Institute.

## Author contributions

**Conceptualization:** Bernd Degen, Niels A. Müller.

**Data curation:** Bernd Degen, Niels A. Müller, Yulai Yanbaev.

**Formal analysis:** Bernd Degen.

**Funding acquisition:** Bernd Degen, Niels A. Müller, Yulai Yanbaev.

**Investigation:** Bernd Degen, Niels A. Müller, Yulai Yanbaev.

**Methodology:** Niels A. Müller, Yulai Yanbaev.

**Software:** Bernd Degen.

**Writing – original draft:** Bernd Degen, Niels A. Müller.

**Writing – review & editing:** Yulai Yanbaev.

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
