## [Decision Letter · Decision Letter 0]

11 Feb 2025

PONE-D-24-56470Machine learning techniques for continuous genetic assignment of geographic origin of forest treesPLOS ONE

Dear Dr. Degen,

Thank you for submitting your manuscript to PLOS ONE. After careful consideration, we feel that it has merit but does not fully meet PLOS ONE’s publication criteria as it currently stands. Therefore, we invite you to submit a revised version of the manuscript that addresses the points raised during the review process.

We look forward to receiving your revised manuscript.

Kind regards,

Maher Maalouf, Ph.D.

Academic Editor

PLOS ONE

Journal Requirements:

“BD & NM: Waldklimafonds WKF-22WC4111 01 & WKF-2219WK60A4 (German Federal Ministry of Food and Agriculture & German Federal German Ministry for the Environment, Nature Conservation and Nuclear Safety);  YY: No 23RL-1F017 from the Higher Education and Science Committee of the Republic of Armenia “

Reviewers' comments:

Reviewer's Responses to Questions

**Comments to the Author**

1. Is the manuscript technically sound, and do the data support the conclusions?

Reviewer #1: Yes

Reviewer #2: Yes

2. Has the statistical analysis been performed appropriately and rigorously? 

Reviewer #1: Yes

Reviewer #2: Yes

3. Have the authors made all data underlying the findings in their manuscript fully available?

Reviewer #1: Yes

Reviewer #2: Yes

4. Is the manuscript presented in an intelligible fashion and written in standard English?

Reviewer #1: Yes

Reviewer #2: Yes

5. Review Comments to the Author

Reviewer #1: The article explores the application of machine learning regression techniques to identify the geographic origin of forest trees using genetic data. It evaluates five methods—Nearest Neighbor (NN), Gaussian Process Regression (GPR), Genomic Prediction (GP), and Deep Learning (DL)—across two datasets: one for European beech and another for pedunculate oak. The study concludes that grid-based GPR and DL demonstrate the highest accuracy, achieving errors below 8% for 90% of the samples. These methods have significant potential for tracking seed sources, detecting mislabeling, combating illegal logging, and enhancing forest management. The article is suitable for publication after minor revisions addressing the following points:

1. The authors should clarify the connection between terms like discrete and continuous assignments and machine learning concepts such as classification and regression.

2. The structure of the data matrix (X) must be clearly described for better understanding.

3. The quality of the figures needs improvement, ensuring the text on them is legible.

Reviewer #2: This is an interesting manuscript attempting to compare methods for assigning individuals within the species range based on individual genotypic data and know the distribution of genetic variants across latitudinal and longitudinal dimensions of the species. The manuscript tests several methods and implements them in two species: Fagus sylvatica and Quercus robur. The two species have different data sets for the number of individuals, the number and types of SNP markers, and the area and density of samples.

The manuscript seems technically sound, and the authors demonstrate good knowledge of the statistical parts of the study. The choice of several different and competing methods is a strong aspect of the paper.

I would expect a comment that the selection of the species beech and oak, both heavy-seeded with limited seed transfer capacity, could be beneficial for this study. Other species, such as conifers, e.g. Scots pine, would probably not allow for such efficient tracking of actual to predicted locations of individuals or groups due to a different spatial structure of genetic diversity.

I appreciate the comments that these methods might be of particular importance in the management of forest genetic resources and controlling the transfer of forest genetic material. However, I would also like to see the comments on how to design sampling data for possible future studies aimed at capturing spatial patterns of genetic diversity, which might be used to predict individual or group locations. In particular, I’m interested in the opinion of whether individual-based sampling or population sampling seems better for this type of research.

Overall, the manuscript is well written and has the potential to be influential for others interested in using genetic data to capture spatial patterns of genetic diversity distributions over larger geographic scales. I fully recommend this manuscript for publishing in PLOS-ONE; however, I provide some comments below, which might be helpful for further improvement of the manuscript.

Minor comments:

111, how many SNPs were nuclear vs. chloroplast? Chloroplast SNPs could be falsified due to the presence of chloroplast sequences in the nuclear genome (NUPTs –nuclear-encoded chloroplast DNA). How the chloroplast SNPs were verified? Based on coverage?

A comment about the usefulness of nuclear or chloroplast markers for spatial analyses is needed. Chloroplast markers are well known in oaks to be maternally inherited and

114, how was the admixture level determined?

127 ‘missing values’

132, how many were chloroplast vs nuclear SNPs ?

143, how were the nuclear and chloroplast SNPs used to obtain joint kinship matric? Uniparental and bi-parental markers have different implications for kinship construction. Were chloroplast SNPs threated as homozygous?

170, what is the rationale for 90:10 proportion of training and testing datasets?

TABLES, consider standardizing the order of methods. Now, the Table 1 has a different order of methods compared to other tables.

Table 3 and Table 4 are mentioned that they are for beech. It seems that Table 4 is for oak. ??

In my opinion, the figures 5-8 presenting outliers, should be presented in supplementary materials.

304, 10 group outliers out of how many? 99 groups?

321, 18 group outliers out of 188? (188 are mentioned earlier as locations, not as groups)

The citation format for references needs to conform to the journal standards.

6. PLOS authors have the option to publish the peer review history of their article (what does this mean? ). If published, this will include your full peer review and any attached files.

**Do you want your identity to be public for this peer review?** For information about this choice, including consent withdrawal, please see our Privacy Policy .

Reviewer #1: No

Reviewer #2: No

---

## [Author Response · Author response to Decision Letter 1]

4 Apr 2025

Reviewer 1

The authors should clarify the connection between terms like discrete and continuous assignments and machine learning concepts such as classification and regression.

Yes, clarified in the text. (L.64-66: “Spatial-genetic assignment methods can be subdivided into discrete and continuous approaches. The distinction aligns closely with the fundamental machine learning terms—classification and regression, respectively.”)

The structure of the data matrix (X) must be clearly described for better understanding.

Yes, a paragraph on the data matrix (X) has been added. (L. 131-136: “The data matrix X represents the genetic information used for the geographic assignment.

It consists of rows corresponding to individuals and columns corresponding to single nucleotide polymorphisms (SNPs). Each element in the matrix encodes the genotype of an individual at a given SNP locus. Rows (𝑛): Each row represents a single tree individual (𝑖=1, 2,…,𝑛), where 𝑛 n is the total number of sampled individuals. Columns (𝑝): Each column corresponds to a specific SNP marker (j=1,2,…,p), where 𝑝 is the total number of SNPs included in the analysis.”)

The quality of the figures needs improvement, ensuring the text on them is legible.

The low quality of the figures is caused by the pdf-creation step of the editorial management system. The original quality of the uploaded files is good.

Reviewer 2

I would expect a comment that the selection of the species beech and oak, both heavy-seeded with limited seed transfer capacity, could be beneficial for this study. Other species, such as conifers, e.g. Scots pine, would probably not allow for such efficient tracking of actual to predicted locations of individuals or groups due to a different spatial structure of genetic diversity.

We have added a phrase on that in the discussion chapter.

I appreciate the comments that these methods might be of particular importance in the management of forest genetic resources and controlling the transfer of forest genetic material. However, I would also like to see the comments on how to design sampling data for possible future studies aimed at capturing spatial patterns of genetic diversity, which might be used to predict individual or group locations. In particular, I’m interested in the opinion of whether individual-based sampling or population sampling seems better for this type of research.

We covered this point in the discussion.

Minor comments:

111, how many SNPs were nuclear vs. chloroplast? Chloroplast SNPs could be falsified due to the presence of chloroplast sequences in the nuclear genome (NUPTs –nuclear-encoded chloroplast DNA). How the chloroplast SNPs were verified? Based on coverage?

We have clarified the number of nuclear and plastid SNPs and the development strategy for the plastid markers in the text.

A comment about the usefulness of nuclear or chloroplast markers for spatial analyses is needed. Chloroplast markers are well known in oaks to be maternally inherited and

114, how was the admixture level determined?

We have added text in that.

127 ‘missing values’

Has been corrected!

132, how many were chloroplast vs nuclear SNPs ?

Has been added to the text (220 in total: 199 nSNPs, 16 cpSNPs, 5 mtSNPs)

143, how were the nuclear and chloroplast SNPs used to obtain joint kinship matric? Uniparental and bi-parental markers have different implications for kinship construction. Were chloroplast SNPs threated as homozygous?

Yes, the cp-SNPs and mt-SNPs were treated as homozygotes. This information has been added to the text.

170, what is the rationale for 90:10 proportion of training and testing datasets?

This cross-validation with a splitting of 90% training data and 10% test data and 5 repetitions was done to find the best hyper-parameter configuration. This were default values. For the final application of all methods, a leave-one-out approach was applied. This means all individuals except one were used for model training (with the before selected optimal hyper-parameter configuration) and the prediction of the longitude and latitude of the test individual.

TABLES, consider standardizing the order of methods. Now, the Table 1 has a different order of methods compared to other tables.

Good point! The tables have been unified.

Table 3 and Table 4 are mentioned that they are for beech. It seems that Table 4 is for oak. ??

Yes, table 4 is for the oak data => has been corrected

In my opinion, the figures 5-8 presenting outliers, should be presented in supplementary materials.

Yes, we put them into “supplementary_1.pdf”

304, 10 group outliers out of how many? 99 groups?

Yes, has been added.

321, 18 group outliers out of 188? (188 are mentioned earlier as locations, not as groups)

Yes, has been added

The citation format for references needs to conform to the journal standards.

Yes, has been checked and corrected

---

## [Decision Letter · Decision Letter 1]

5 May 2025

Machine learning techniques for continuous genetic assignment of geographic origin of forest trees

PONE-D-24-56470R1

Dear Dr. Degen,

We’re pleased to inform you that your manuscript has been judged scientifically suitable for publication and will be formally accepted for publication once it meets all outstanding technical requirements.

Kind regards,

Maher Maalouf, Ph.D.

Academic Editor

PLOS ONE

Additional Editor Comments (optional):

Reviewers' comments:

Reviewer's Responses to Questions

**Comments to the Author**

1. If the authors have adequately addressed your comments raised in a previous round of review and you feel that this manuscript is now acceptable for publication, you may indicate that here to bypass the “Comments to the Author” section, enter your conflict of interest statement in the “Confidential to Editor” section, and submit your "Accept" recommendation.

Reviewer #1: All comments have been addressed

Reviewer #2: All comments have been addressed

2. Is the manuscript technically sound, and do the data support the conclusions?

Reviewer #1: Yes

Reviewer #2: Yes

3. Has the statistical analysis been performed appropriately and rigorously? 

Reviewer #1: Yes

Reviewer #2: Yes

4. Have the authors made all data underlying the findings in their manuscript fully available?

Reviewer #1: Yes

Reviewer #2: Yes

5. Is the manuscript presented in an intelligible fashion and written in standard English?

Reviewer #1: Yes

Reviewer #2: Yes

6. Review Comments to the Author

Reviewer #1: The authors have thoroughly addressed my previous comments and incorporated the necessary modifications into the revised manuscript, resulting in a clearer and more comprehensive presentation of their work.

Reviewer #2: The Authors did a good job responding to all points and questions I raised in the former review report. In particular, I like the discussion of population vs. individual-based-sampling for assessing genetic diversity of the species, in particular important in the context of the topic of the manuscript. I believe, the manuscript reached the state ready for publication

7. PLOS authors have the option to publish the peer review history of their article (what does this mean? ). If published, this will include your full peer review and any attached files.

**Do you want your identity to be public for this peer review?** For information about this choice, including consent withdrawal, please see our Privacy Policy .

Reviewer #1: No

Reviewer #2: No

---

## [Editor Report · Acceptance letter]

PONE-D-24-56470R1

PLOS ONE

Dear Dr. Degen,

I'm pleased to inform you that your manuscript has been deemed suitable for publication in PLOS ONE. Congratulations! Your manuscript is now being handed over to our production team.

Kind regards,

on behalf of

Dr. Maher Maalouf

Academic Editor

PLOS ONE